# Tripartite: Tackling Realistic Noisy Labels with More Precise Partitions

**DOI:** 10.3390/s25113369

**Published:** 2025-05-27

**Authors:** Lida Yu, Xuefeng Liang, Chang Cao, Longshan Yao, Xingyu Liu

**Affiliations:** 1School of Arts and Sciences, Beijing Normal University, Beijing 100875, China; 202211079020@mail.bnu.edu.cn; 2School of Artificial Intelligence, Xidian University, Xi’an 710071, China; yaolongshan@stu.xidian.edu.cn; 3Guangzhou Institute of Technology, Xidian University, Guangzhou 510555, China; caochang@stu.xidian.edu.cn (C.C.); xingyuliu@stu.xidian.edu.cn (X.L.)

**Keywords:** realistic noisy label, uncertain samples, tripartition, bipartition

## Abstract

Samples in large-scale datasets may be mislabeled for various reasons, and deep models are inclined to over-fit some noisy samples using conventional training procedures. The key solution is to alleviate the harm of these noisy labels. Many existing methods try to divide training data into clean and noisy subsets in terms of loss values. We observe that a reason hindering the better performance of deep models is the uncertain samples, which have relatively small losses and often appear in real-world datasets. Due to small losses, many uncertain noisy samples are divided into the clean subset and then degrade models’ performance. Instead, we propose a Tripartite solution to partition training data into three subsets, *uncertain*, *clean* and *noisy* according to the following criteria: the inconsistency of the predictions of two networks and the given labels. Tripartite considerably improves the quality of the clean subset. Moreover, to maximize the value of clean samples in the uncertain subset and minimize the harm of noisy labels, we apply low-weight learning and a semi-supervised learning, respectively. Extensive experiments demonstrate that Tripartite can filter out noisy samples more precisely and outperforms most state-of-the-art methods on four benchmark datasets and especially real-world datasets.

## 1. Introduction

The success of DNNs in AI-based computer vision systems—such as autonomous vehicles and surveillance networks—relies heavily on large-scale datasets with precise human annotations. However, acquiring high-quality labeled data for sensor-driven applications is prohibitively costly and time-consuming. To address this, practitioners often resort to cost-efficient alternatives such as web-crawling [1], online queries [2,3,4], crowdsourcing [5,6], and so on. Unfortunately, these methods introduce label noise due to sensor-specific challenges: motion-blurred LiDAR scans may be mislabeled as static objects, or occluded pedestrians in camera footage might be ambiguously annotated. Studies [7,8,9,10] demonstrate that DNNs can easily over-fit to noise, which significantly degrades their generalization ability, severely degrading their ability to generalize to real-world scenarios—for instance, failing to recognize partially obscured traffic signs or misclassifying low-resolution objects in edge-device deployments [11,12].

There have been many efforts [13,14,15] to tackle noisy labels. Most of them reach a consensus that the key is to alleviate the impact of noisy labels for network training. Loss-correction-based methods [16,17,18,19] aim to rectify the losses of noisy samples during training but may mistakenly rectify some clean samples. Sample-selection based methods [20,21,22,23] tackle noisy labels by partitioning the training data into the clean and noisy subsets separately, then using them for network training in different ways. These bipartition methods commonly utilize the following partition criteria: (1) The *Small-loss criterion*  [20] assumes that the losses of noisy samples are significantly greater than those of the clean samples during training. Therefore, they try to find a threshold, *T*, and select the samples, whose losses <T, as clean samples. The others are treated as noisy samples. (2) The *Gaussian Mixture Model* (GMM) *criterion*  [21] believes that the statistical distribution of noisy samples’ losses is different from that of clean samples’ losses. It aims to find the probability of a sample being noisy or clean by fitting a mixture model. (3) The *consistency criterion*  [23,24,25] assumes that noisy samples exhibit inconsistencies in predictions or inconsistencies in label distributions when compared to their neighboring samples. Based on this, the samples are divided into clean and noisy subsets.

However, in the early stage of training, the network’s capacity is limited, posing a considerable challenge of identifying some samples as clean or noisy. We refer to samples that are difficult to correctly classify as either clean or noisy as uncertain samples. Our observations indicate that a substantial cohort of uncertain samples are distributed close to the classification boundaries, e.g., samples grouped by the red curve shown in Figure 1a, especially in real-world datasets. Given that uncertain samples encompass both noisy and clean instances, bipartition methods may misclassify some noisy uncertain samples as clean. Subsequently, as vision systems fit these samples during training, they may result in smaller losses, as illustrated in the green area between the two red dashed lines in Figure 1b. These samples are very likely to be included in the clean subset again and then be over-fitted by the vision system during later training phases. This over-fitting can lead to performance degradation in the vision system, particularly in applications where robustness to label noise is critical. In the context of AI-based computer vision sensors and systems, the challenge of handling uncertain samples becomes even more pronounced.

In this case, we face a challenging issue: loss-based methods cannot accurately divide samples when some noisy samples also exhibit smaller losses. This is particularly problematic in AI-based computer vision systems, where uncertain samples are often distributed close to the classification boundaries and are prone to being misclassified into either the clean or the noisy subset. As a result, traditional bipartition methods struggle to handle uncertain samples effectively, which can lead to performance degradation in vision systems, especially in applications requiring high robustness to label noise. To address these challenges, we propose a novel method, Tripartite, which divides training data into three subsets: clean, noisy, and uncertain. This approach aims to improve the quality of the clean subset by leveraging the consistencies of predictions from two different networks and the given label. In the context of computer vision sensors and systems, this method can enhance the reliability of the system by reducing the impact of label noise on the training process. In the Tripartite method, samples in the uncertain subset typically exhibit lower classification confidence from the networks, meaning that the two networks are likely to give inconsistent predictions. Conversely, samples with higher classification confidence, where the two networks provide consistent predictions, are assigned to either the clean or the noisy subset. If the prediction matches the given label, the sample is placed in the clean subset; if it differs, it is placed in the noisy subset. To effectively utilize the data, we apply a low-weight training strategy for samples in the uncertain subset, a semi-supervised training strategy for samples in the noisy subset using pseudo labels instead of given labels, and a traditional supervised training strategy for samples in the clean subset.

Unlike bipartition methods, which may misclassify noisy, uncertain samples into the clean subset, Tripartite enhances the quality of the clean subset by relocating such samples into the uncertain subset. This approach is specifically designed to minimize the impact of noisy labels within the clean subset and maximize the utility of clean uncertain samples within the uncertain subset, which is crucial for maintaining the robustness and accuracy of AI-based computer vision systems in real-world applications. Extensive experiments demonstrate that Tripartite outperforms state-of-the-art (SOTA) methods in handling label noise, making it particularly suitable for vision sensors and systems that require high reliability.

In particular, sensor-based data systems—such as wearables, autonomous platforms, or IoT devices—frequently suffer from ambiguous annotations due to noise from environmental conditions, hardware limitations, or manual labeling errors. These challenges often lead to boundary samples or class confusion in readings (e.g., transition between similar motion states), which are difficult to handle using conventional clean/noisy bipartition schemes. The Tripartite framework is well suited for such sensor environments. By explicitly introducing an “uncertain” subset and applying low-weight and semi-supervised strategies, Tripartite effectively isolates ambiguous samples and enhances training robustness. Moreover, the proposed realistic noise generation approach mimics the semantic confusion patterns often found in sensor data, making the framework adaptable to multimodal, dynamic, or edge-deployed AI systems.

The key contributions of our work are threefold:We propose a novel partition criterion, which divides training data into three subsets: *uncertain, noisy, and clean.* It alleviates the uncertain sample division problem of bipartition methods and minimizes the harm of noisy labels by improving the quality of the clean subset.We design a low-weight training strategy that aims to maximize the value of clean samples in the uncertain subset while minimizing the influence of potential noise. This strategy is particularly beneficial for vision systems operating in environments with ambiguous or noisy sensor data.To simulate the characteristics of noisy labels in real-world datasets, we design a synthetic class-dependent label noise model on CIFAR datasets, referred to as realistic noise. This model flips labels of samples from two different classes at controlled ratios based on their class similarity, providing a more accurate representation of label noise challenges faced by computer vision systems in practical scenarios.

## 2. Related Work

There have been many studies to address noisy labels. They all try to lower the impact of noisy labels to improve the classification performance of methods during training. To this end, two different ideas were proposed: *loss correction* and *sample selection*.

### 2.1. Loss Correction

The specific methods include *noise transition matrix*  [16,27,28,29], *robust loss functions*  [30,31,32,33,34], *label correction*  [19,35,36], etc. *Noise transition matrix* methods construct the label transition matrix to estimate the possibilities of noisy labels transitioning among multiple classes. F-correction [16] performs forward correction by multiplying the transition matrix with the softmax outputs in the forward propagation. Later, Hendrycks et al. [37] improved the corruption matrix using a small clean dataset. *Robust-loss* methods aim to design loss functions that are robust to noisy labels, such as Mean Absolute Error (MAE) [30]; Improved MAE [32], which is a reweighted MAE; Generalized Cross-Entropy loss (GCE) [33], which is a generalization of MAE; and Symmetric Cross-Entropy [34], which adds a reverse cross-entropy term to the usual cross-entropy loss. *Label correction* methods try to rectify the noisy labels according to the network predictions during the training. Joint optimization [19] learns network parameters and infers the true labels simultaneously. PENCIL [35] adopts label probability distributions to supervise network learning and updates these distributions through back-propagation in each epoch. Inspired by “early learning”, ELR+ [36] uses the network predictions in the early training stage to correct the noisy labels.

The above methods do not divide training data into clean and noisy samples. They may mistakenly rectify the losses of clean samples and introduce new, noisy labels into the training data. Therefore, sample selection methods were proposed.

### 2.2. Sample Selection

These methods try to divide training data into “clean” and “noisy” subsets according to a specific partition criterion. Afterward, they apply different strategies to train the model on two subsets separately. The existing partition criteria are mainly based on the training loss, e.g., the *small-loss criterion*  [20], the *Gaussian Mixture Model* (GMM) *criterion*  [21] and the *consistency-based criterion.* The *small-loss criterion* selects training samples with small loss as clean ones by setting a threshold *T*. In particular, MentorNet [38] pretrains a teacher network to select clean samples to guide the training of a student network. Co-teaching [20] trains two networks where each network selects small-loss samples to feed its peer network for updating parameters. Furthermore, Co-teaching+ [22] emphasizes the help of inconsistency to the network. However, all these methods only use the data with smaller losses for training without considering the data with larger losses. Meanwhile, finding a feasible *T* is very challenging. GMM assumes that the distributions of losses of noisy samples and clean samples follow two normal distributions, respectively. DivideMix [21] employs two networks, each of which selects data for the other in the training and then applies a semi-supervised method to learn the noisy samples. This greatly improves the performance of networks. However, in real-world datasets, the noisy labels often appear between similar classes, such as *whale* and *dolphin,* and *oak_tree* and *maple_tree* in CIFAR-100. GMM may mistake the noisy, uncertain samples and the clean, uncertain samples because of the small difference between their losses. This leads to a low quality of training data partition. TCL [39] presents a novel twin contrastive learning model, which constructs a GMM model over the representations and detects the examples with wrong labels by another two-component GMM and then infers the posterior probability of one example having clean labels through the two-component GMM. It can learn discriminative representations aligned with estimated labels through mixup and contrastive learning, but the posterior probability may not be precise for some uncertain samples. PES [40] finds that the latter layers in a DNN are much more sensitive to label noises, so it separates a DNN into different parts and progressively trains them. It selects a training sample into the clean subset or the noisy subset according to whether the network prediction is consistent with the given label and then applies a semi-supervised learning on the noisy subset. DLT [41] proposes a novel noisy label detection method that records the loss value of each sample and calculates dynamic loss thresholds, then uses these thresholds to detect noisy labels. These two methods may mistake the noisy, uncertain samples and the clear, uncertain samples as well. The *consistency-based criterion* relies on the inconsistency in predictions or label distributions of noisy samples to select data. JoCoR [23] emphasizes the consistency of the two networks. TC-Net [24] constrains the transform consistency between the original images and the transformed images for network training. NCE [25] determines whether a candidate is a noisy sample by estimating the inconsistency in the label distribution between the candidate sample and its contrastive neighbors. These methods still partition samples into clean and noisy subsets, thereby risking confusion between noisy, uncertain samples and clean, uncertain samples. To address the web label noise, DSOS [42] is proposed to divide data into clean, in-vocabulary noise, and out-of-vocabulary noise subsets. It reported the best performance among SOTA methods on web-crawled datasets without considering uncertain samples. By contrast, we propose the Tripartition criterion, which divides training data into clean, uncertain, and noisy subsets. A more precise partition can effectively lower the harm of noisy labels for network training. More recent efforts have also explored end-to-end frameworks to address the challenges of noisy labels. For instance, SplitNet [43] introduces a learnable clean–noisy partitioning strategy that dynamically assigns training samples into subsets based on representation-level features. It enables more adaptive filtering during training without relying on static thresholds. Additionally, AEON [44] tackles instance-dependent and out-of-distribution noise through an adaptive estimation mechanism that separates the noise types and applies targeted training objectives. These approaches further emphasize the importance of noise-aware partitioning and provide complementary perspectives to our proposed Tripartition strategy.

## 3. The Design of Realistic Label Noise

In real-world noisy datasets, noisy labels often come from similar classes. We observe that many noisy labels are introduced into the aforementioned uncertain data. However, symmetric noise, which randomly replaces labels with other labels, ignores inter-class similarities. Thus, while many existing methods achieve remarkable performance on such types of label noise, they may not perform well on real-world noisy datasets. Recent studies have increasingly addressed label noise across different domains. For example, Liu et al. [45] proposed a discriminative approximate regression projection method to enhance robust feature extraction. Wang et al. [46] adopted a meta-learning framework to optimize lifetime prediction under label noise in battery data. Zhang et al. [47] tackled label noise in cross-modal person re-identification using deep learning. These methods emphasize robustness in feature learning or task adaptation, which complements our focus on partitioning strategies under noisy supervision. Therefore, we propose a new noise label on CIFAR datasets called *realistic noise* to simulate real-world noise better. It is used to evaluate the performance of all methods and explain the mechanism of Tripartite. It flips labels in class pairs at a certain ratio, which is proportional to the similarity between two classes. The more similar the two classes are, the more noisy labels are introduced. Although the asymmetric noise in previous works also attempts to mimic the structure of real-world label noise, the assigned classes are often defined based on intuitive judgments rather than the nature of the data. In contrast, our proposed realistic noise calculates the similarity between classes and transfers noisy labels, better simulating the noise in the real world. The detailed procedure for generating the noise transition matrix of realistic noise is shown below.

**Step 1.** We first train a ResNet50 on CIFAR-10 and CIFAR-100 until it converges. The trained model achieves 94.67% test accuracy on CIFAR-10 and 78.85% on CIFAR-100. We then consider the weight vector between the last fully connected layer and a node (a class) in the output layer as the prototype of the class [48].

**Step 2.** We pair all prototypes into *N* class pairs $\left(c_{i},c_{j}\right)$ i≠j, then compute their cosine similarities and sort them in descending order. We select the top *K* pairs and flip some labels in each pair, where *K* is set to 10 for CIFAR-10 (Table 1) and 60 for CIFAR-100 (Table 2).

**Step 3.** The *K* pairs are partitioned into three similarity levels. Each level has a weight *w*, which is proportional to the similarity between two classes in the pair. The higher *w* is, the more labels will be flipped. *w* is selected from {0.9,0.6,0.3} for CIFAR-100, and {0.9,0.8,0.7} for CIFAR-10. As more similar pairs are selected from CIFAR-100, the weight step between different levels becomes larger. If the similarity of a class pair is not in the top *k* pairs, the *w* is set as 0.

**Step 4.** To construct the noise transition matrix, we temporarily fill in the positions of the top *K* class pairs with the corresponding weights $w^{i,j}$, where *i* and *j* denote the i−th row and j−th column in the matrix. If a row is empty, we will fill the non-diagonal position uniformly with 1. Other positions are filled in with 0. In order to make the sum of all weights in each row to be 1, the $w^{i,j}$ is replaced by(1)$${\hat{w}}^{i,j}=\frac{w^{i,j}}{\sum_{j=1}^{n}w^{i,j}},$$
where *n* is the length of columns.

**Step 5.** Given a noise ratio *r*, we fill in the diagonal of the matrix with (1−r) and multiply the other elements by *r*.

Figure 2 shows the generated noise transition matrices of CIFAR-10 with noise ratios of 20% and 50%, respectively.

Figure 3 visualizes the feature distributions of three similar class-pairs in CIFAR-100 using t-SNE. We train a ResNet50 for 200 epochs on CIFAR-100 without noisy labels and use the output of the last convolutional layer as the feature of each sample. We can see that many uncertain samples are distributed around the classification boundary between two similar classes.

## 4. The Proposed Method

To explain the mechanism of Tripartite, we first analyze the distributions of uncertain, clean, and noisy samples and then derive the logic of the data partition in Section 4.2. Section 4.3 details the proposed Tripartition criterion. Section 4.4 presents the training strategies for three subsets.

### 4.1. Preliminaries

In our method, we train two networks, denoted by $f\left(\theta_{k}\right),k=\left\{1,2\right\}$, which are initialized with different weights. We assume that they can learn not only the reliable information from training data but also some different views, which may result in the inconsistent predictions of networks on uncertain samples. $P_{k}$ is the predicted label of $f(x;\theta_{k})$ on an input *x*. The proposed methods mentioned hereinafter are all based on the predictions of these two networks.

Compared to DISC [50], which uses prediction consistency for binary partitioning, Tripartite introduces a third “uncertain” subset and applies subset-specific learning strategies. This improves robustness under ambiguous or instance-dependent noise, where binary categorization may be insufficient.

### 4.2. Distribution of Training Data and the Logic of Tripartite

For an understanding of the noisy label problem on realistic datasets, we visualize the feature distribution of samples from the realistic CIFAR-10 in Figure 1a, where each color represents a class, and a black dot denotes a noisy sample. One can see that each class has a clear cluster. There are also many samples distributed between clusters; e.g., the dots grouped by the red curve. These are the aforementioned uncertain samples, which have some shared features of two or even more classes. It is very challenging for networks to distinguish them as clean or noisy samples. We also observe a small portion of uncertain samples that are very special. As two networks in our method could learn different views from a sample, these special samples may be distributed close to the classification boundary in the feature space of one network, but distribute in the clusters in the feature space of another network. The detailed discussion is given in Section 5.4.

To simplify the discussion, we analyze the distribution in the feature space of one network. Let us take the categories *cat* and *dog* as an example, where 50 samples are uniformly selected from each category for a clearer visualization, as shown in Figure 4. The samples on the outside of the bounding box in Figure 4b are the certain samples. They belong to two cases. The first is *certain samples with clean labels*, which are red dots and blue triangles. As the label information is consistent with the sample features, networks will fit these samples in the early training stage. Moreover, they are distant from the classification boundary, so networks can correctly recognize them. The second is *certain samples with noisy labels*, which are blue dots, red triangles, and black stars. As the information provided by the given labels is inconsistent with the features of these samples, networks would not fit them in the early training stage. These samples could also be correctly predicted by both networks, but the prediction is obviously inconsistent with the given label.

The samples in the bounding box in Figure 4b are considered as the uncertain samples. They belong to three cases. The first is *uncertain samples with clean labels*, which are red dots and blue triangles. Although the label information is correlated with the sample features, they have some shared features between the similar categories due to being distributed close to the classification boundary. The second is *uncertain samples that are mislabeled into similar categories*, which are blue dots and red triangles. They also have some shared features between the similar categories for the same reason. The third is *uncertain samples that are mislabeled into dissimilar categories*, which are black stars. Their features are irrelevant to the information provided by the given labels, so their labels are similar to the symmetric noise, which are transferred to dissimilar categories. As many samples in the aforementioned three cases are distributed close to the classification boundary and have shared features of two categories, two networks in our method are likely to give inconsistent predictions.

### 4.3. Tripartition Method and Criteria

For the realistic label noise, noisy labels tend to happen between similar categories. Our investigation shows that networks with varied initialization may learn different views from a sample. When two networks give inconsistent predictions, it indicates their classification confidences are low. Then, we consider such samples as uncertain samples. When two networks give a consistent prediction, their classification confidences are usually high. These samples are considered as certain samples and then divided into the clean or noisy subset according to the consistency between the prediction and the given label. The logic of Tripartite is shown in Figure 5. The detailed partition criteria are given below.

#### 4.3.1. The Selection Criterion for the Uncertain Subset

Let $p_{1}$ and $p_{2}$ denote the predictions of the two networks $f(\theta_{1})$ and $f(\theta_{2})$, respectively.(2)$$\begin{array}{c} P_{1}\neq P_{2} \end{array}$$

Samples in the uncertain subset mainly have clean or noisy labels of similar classes. Moreover, some uncertain samples have noisy labels of dissimilar classes. As many uncertain samples are distributed close to the classification boundary, different networks likely give inconsistent predictions on such samples, especially in the early training stage. We then apply the inconsistent predictions of two networks, Equation (Equation 2), to select them. Our experiments demonstrate that the population of these data will gradually decrease with the increase in the classification confidence of networks.

Although this inconsistency arises from the predictions of two independently initialized networks, such an initialization difference is an intentional design choice (see Section 4.1). It allows the two networks to learn different feature perspectives, which helps identify ambiguous or boundary samples more effectively. As shown in the visualization in Section 5.4, these uncertain samples often appear inconsistent in one network while clustered in another, validating the effectiveness and stability of this criterion despite initialization randomness.

#### 4.3.2. The Selection Criterion for the Noisy Subset

(3)$$\begin{array}{c} P_{1}=P_{2}\neq GL \end{array}$$
The noisy subset mainly includes certain samples with noisy labels. As the predictions of two networks are consistent and different from the given label, the sample is very likely to be a noisy sample. Meanwhile, the networks will not fit them in the early training stage and can provide a high-confidence prediction; we then apply this inconsistency, Equation (Equation 3), to select them.

#### 4.3.3. The Selection Criterion for the Clean Subset


(4)
$$\begin{array}{c} P_{1}=GL\cap P_{2}=GL \end{array}$$


The clean subset mainly includes certain samples with clean labels. As these samples are distributed far from the classification boundary and their features and the label information are rather correlated, networks can learn the mapping between data features and labels in the early training stage. So the predictions of the two networks and the given label should be consistent. We apply this consistency, Equation (Equation 4), to select clean samples.

Our realistic noise generation strategy is conceptually related to instance-dependent label noise simulation methods, such as RoG [51]. However, there are fundamental differences. RoG generates noise by first training a deep discriminative model and then fitting a generative classifier (e.g., Gaussian) over the learned feature representations, and finally identifying low-likelihood instances—via the Mahalanobis distance—to relabel them. This approach is highly model-dependent and relies on instance-level statistical estimation over the learned feature space.

In contrast, our method constructs a class-level confusion matrix based on cosine similarity between class prototypes or semantic embeddings. It introduces label noise by probabilistically flipping labels between semantically similar classes without requiring any model training or feature inference. This makes our method simpler, model-agnostic, and more scalable.

### 4.4. Learning Strategies for the Three Subsets

We design different learning strategies for three subsets, respectively. Let us consider a classification problem with *C* classes. The training set consists of *n* examples, where $\left\{(x^{i},y^{i})\right\}$ is the $i^{th}$ input sample and $y^{i}\in {\left\{0,1\right\}}^{C}$ is a one-hot label of *C* dimensions vector corresponding to the class of $x^{i}$. The network maps the input $x^{i}$ into a *C*-dimensional encoding and feeds it into a softmax function to estimate the probability $p^{i}$ of $x^{i}$ for each class.

#### 4.4.1. The Learning Strategy for the Uncertain Subset

The uncertain subset includes samples with either clean labels or noisy labels. We wish to utilize the valuable information of clean samples to improve the discriminative ability of networks, while lowering the harm of noisy labels in this subset. Therefore, we apply a low-weight cross-entropy loss for the network training.(5)$$Loss_{u}=\lambda_{u}\left(-\frac{1}{n}\sum_{i=1}^{n}\sum_{c=1}^{C}y_{c}^{i}log\left(p_{c}^{i}\right)\right),$$
where $\lambda_{u}$ is a weight in the range (0, 1), $p_{c}^{i}$ is the estimated probability of $x^{i}$ being in class *c*.

#### 4.4.2. The Learning Strategy for the Noisy Subset

To lower the impact of noisy labels, it would be better to reassign a reasonable label for noisy samples. To this end, we utilize the ensemble of two networks’ predictions to “co-guess” labels. This strategy is similar to the method in DivideMix [21], which had reported a reliable “guess”.(6)$${\hat{q}}^{i}=\frac{1}{2M}\sum_{m=1}^{M}\left(f\left({\hat{x}}_{m}^{i},\theta_{1}\right)+f\left({\hat{x}}_{m}^{i},\theta_{2}\right)\right),$$
(7)$$q^{i}={\hat{q}}_{c}^{i\frac{1}{T}}/\sum_{c=1}^{C}{\hat{q}}_{c}^{i\frac{1}{T}},\;\mathrm{where}\;c=1,2,\ldots ,C.$$

Given a noisy labeled sample $x^{i}$, we obtain its *M* augmentations $\left\{{\hat{x}}_{1}^{i},\ldots ,{\hat{x}}_{M}^{i}\right\}$ randomly by rotation, flip, cropping, desaturation, contrast, blurring, mixup [52], etc. A new label ${\hat{q}}^{i}$ is guessed by Equation (Equation 6) before each epoch, where $\theta_{1}$ and $\theta_{2}$ are the parameters of two networks, respectively. Then, the guessed label $q^{i}$ is sharpened by Equation (Equation 7), where ${\hat{q}}_{c}^{i}$ is the guessed probability for class *c* of sample $x^{i}$ and *T* is the sharpening temperature. We use the CE loss to train each network.(8)$$Loss_{n}=\lambda_{n}\left(-\frac{1}{n}\sum_{i=1}^{n}\sum_{c=1}^{C}q_{c}^{i}\log\left(p_{c}^{i}\right)\right),$$
where $p^{i}$ is the label probability of $x^{i}$ predicted by the $k^{th}$ network during training, $p^{i}=f\left(x^{i},\theta_{k}\right)$ and k={1,2}, $q^{i}$ is the guessed label probability done by two networks, and $\lambda_{n}$ is a parameter to control the contribution of the noisy subset.

#### 4.4.3. The Learning Strategy for the Clean Subset

As Tripartite ensures the high quality of the clean subset, we apply the cross-entropy loss for the network training,(9)$$Loss_{c}=-\frac{1}{n}\sum_{i=1}^{n}\sum_{c=1}^{C}y_{c}^{i}log\left(p_{c}^{i}\right).$$

Finally, the total loss is(10)$$Loss_{tri}=Loss_{c}+Loss_{n}+Loss_{u}.$$

### 4.5. Pseudo-Code

The pseudo-code of Tripartite is shown in Algorithm 1.
**Algorithm 1:** Tripartite
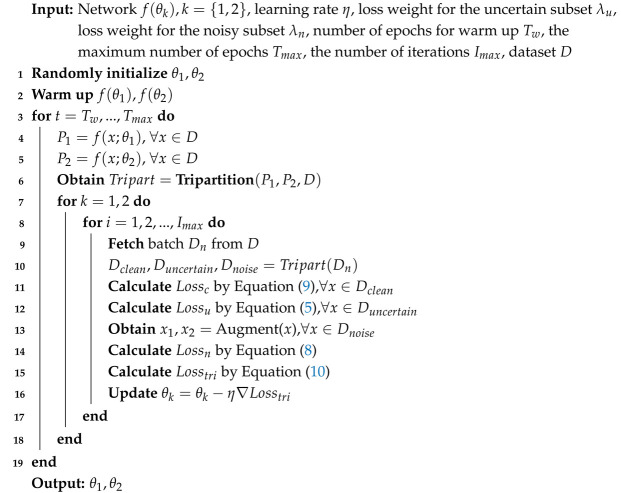


## 5. Experiments

### 5.1. Datasets and Noise Types

We verify the effectiveness of our Tripartite extensively on two datasets with artificial label noises (CIFAR-10 [53], CIFAR-100 [53]) and two real-world datasets (Clothing1M [54], WebVision1.0 [4]).

Both CIFAR-10 and CIFAR-100 contain 60,000 images of size 32 × 32. CIFAR-10 includes 10 super-classes. CIFAR-100 has 100 subclasses. Three different label noises are set on them. (1) *Symmetric noise*: Labels of a certain percentage of training data are flipped to other labels randomly and uniformly. (2) *Instance-dependent noise*: The noisy labels are generated by the method proposed in [55]. (3) *Realistic noise*: The noisy labels are generated based on the similarities between classes introduced in Section 3.

Clothing1M consists of 1 million training images collected from online shopping websites with labels generated according to the surrounding text. Its noise ratio is about 38.5%.

WebVision1.0 is a large-scale dataset with real-world noisy labels. It contains more than 2.4 million images crawled from the web using the 1000 concepts in ImageNet ILSVRC2012 [56]. Its noise ratio is about 20%. For ease of comparison with competing methods, we follow the previous work [57] and then perform the test on a subset comprising the first 50 classes crawled from Google image. The details of these six datasets are listed in Table 3.

### 5.2. Models and Parameters

Following previous works [7,21,36,40,42], we train the 18-layer PreAct Resnet [26] on CIFAR-10 and CIFAR-100 using the SGD optimizer with a learning rate of 0.02, a momentum of 0.9, and a weight-decay of 0.0005 for 300 epochs. The batch size is set to 128. The architectures of the two networks used in our method are the same. However, the networks are randomly initialized with different weights and are fed different randomly augmented data each time, ensuring their differences in the early stages of training. The learning rate is reduced by a factor of 10 after 150 and 200 epochs. We set the warm-up as 10 epochs for CIFAR-10 with all label noises, 30 epochs for CIFAR-100 with symmetric noise and instance-dependent noise, and 20 epochs for CIFAR-100 with realistic noise. The most critical parameters in Tripartite are $\lambda_{n}$ and $\lambda_{u}$. The $\lambda_{n}$ is selected from {0.2, 0.3, 0.4, 0.5, 0.6}, $\lambda_{u}$ is 0.3 on both CIFAR datasets.

For real-world datasets, the experiment settings and parameters are the same as CIFAR datasets except training epochs, learning rate, and network architectures. The batch size is set to 64. For Webvision1.0, we set the warm-up to 20 epochs. We train the network for 100 epochs, and the learning rate is reduced by a factor of 10 after 60 and 100 epochs. For Clothing1M, we set the warm-up to five epochs. We train the network for 60 epochs, and the learning rate is reduced by a factor of 10 after 20 and 40 epochs. Resnet-50 [58], pre-trained on ImageNet (following previous work [7]), is for Clothing1M, and inception-resnet v2 [59] is for WebVsion1.0. The $\lambda_{u}$ is 0.6 for Webvision1.0 and 0.2 for Clothing1M.

The values of $\lambda_{n}$ and $\lambda_{u}$ were determined empirically. For CIFAR datasets, $\lambda_{n}$ was chosen from the candidate set 0.2, 0.3, 0.4, 0.5, 0.6, and $\lambda_{u}$ was fixed at 0.3, which provided stable performance across different noise types. For real-world datasets, these parameters were tuned separately based on experimental validation.

### 5.3. Comparison with SOTA Methods

We compared our method with the baseline Standard CE (the backbone + Cross Entropy), and the recent SOTA methods including Bootstrap [18], F-correction [16], P-correction (PENCIL) [35], Meta-Learning [7], M-correction [60], Co-teaching [20], JoCoR [23], DivideMix [21], ELR+ [36], Co-learning [61], DSOS [42], JNPL [62], PES [40], TCL [39], and DLT [41]. For a fair comparison, we followed previous works and report the combined predictions of two networks on the test set. To ensure statistical reliability, we conducted the experiments five times and report the mean results. Since Tripartite aims at real-world noisy data, in which the noise ratio over 50% is unlikely to happen [61], we focus on the data with the noise ratio less than 50%.

#### 5.3.1. Results on CIFAR-10 and CIFAR-100

Table 4 shows the test accuracy on CIFAR-10 and CIFAR-100 with symmetric noise. We can see that Tripartite outperforms competing methods on CIFAR-100 with all noise ratios and on CIFAR-10 with a 20% noise ratio. Table 5 shows the test accuracy on CIFAR-10 and CIFAR-100 with instance-dependent noise. Tripartite outperforms competing methods at all settings. Table 6 shows the test accuracy on CIFAR-10 and CIFAR-100 with realistic noise. Tripartite achieves a significant performance gain compared to other methods across all noise ratios.

#### 5.3.2. Results on Real-World Datasets

Table 7 and Table 8 list the results on real-world noisy datasets, Clothing1M and WebVision1.0, respectively. We also evaluate the generalization ability of these methods on ImageNet ILSVRC12. Tripartite consistently outperforms competing methods across all large-scale datasets. One can see the performance gains are not very significant on Clothing1M. The possible reason is that some images contain more than one subject. For example, images in Clothing1M are usually full-body shots that include top and bottom clothing but are labeled as either of them. In this case, the networks have difficulty learning the discriminative feature. Hence, Tripartite may partition data with some uncertainty.

As WebVision has more categories and more diverse categories, it allows Tripartite to achieve a considerable improvement than other methods. TCL [39] also performs well on WebVision and ILSVRC12 because of the twin contrastive learning structure. Tripartite outperforms it by about 1% across all criteria. Our investigation reveals that the posterior probabilities inferred by TCL are inaccurate for many uncertain samples that are subsequently selected into the clean subset. Conversely, Tripartite exhibits its robustness in selecting uncertain samples, resulting in a high-quality clean subset.

### 5.4. Visualization of Uncertain Samples

Selecting uncertain samples plays a crucial role in improving the quality of the clean subset. We think that this is the key reason that Tripartite can achieve better performance in the above experiments, especially on realistic label noise. It is worth having an insight into these samples to better understand the mechanism of our Tripartite.

Figure 6a shows the distribution of clean samples (color dots) and uncertain samples (black dots) in CIFAR-10 in the feature space of network 1 in the later training stage. One can see that most uncertain samples are distributed close to the classification boundary. Therefore, network 1 has lower confidence to classify them into either of the classes.

Meanwhile, a few black dots are distributed in the clusters. We plot them only in Figure 6b. This indicates that network 1 has higher confidence to classify them, which seems to contradict our definition of uncertain samples. We then visualize their distribution in the feature space of trained network 2 in Figure 6c. One can see that they are almost distributed at the classification boundary again. This means that network 2 has lower confidence to classify them into either of the classes. The possible reason is that two networks have varied initialization; then, network 1 learns a better discriminative feature in the current training stage, but network 2 does not, and vice versa. As two networks learn different features from such samples, it is difficult to classify them precisely. We consider them as uncertain samples as well.

In summary, uncertain samples consist of two cases. (1) The majority are distributed close to the classification boundary in both feature spaces of the two networks. (2) The minority are distributed at the classification boundary in the space of one network but in the clusters in the space of another network. Neither of the two cases allows two networks to give the same prediction. Therefore, we apply the partition criterion (2) in the Section 4.3.1 to select uncertain samples.

### 5.5. The Quality of Training Data Partition

As the quality of the clean subset is the key reason that Tripartite performs better than the competing methods, we test the partition accuracies of clean and noisy subsets on the realistic label noise and the instance-dependent noise. The partition accuracy of the clean subset is defined as(11)$$Partition\;Accuracy=\frac{1}{Num_{c}}\sum_{i=1}^{Num_{c}}I(GL_{i}=TL_{i}),$$
where $Num_{c}$ denotes the number of samples in the clean subset and $GL_{i}$ and $TL_{i}$ are the given label and the ground-truth label of the i−th sample in the clean subset. I(·) is an indicator function, where I(·)=1 if · is true; otherwise, I(·)=0. The partition accuracy of the noisy subset is analogous to Equation (Equation 11).

The comparison is conducted among *Tripartite*, the *small-loss criterion* [20], and the *GMM criterion* [21]. The results are shown in Figure 7. We can see that Tripartite largely outperforms the other two criteria on the clean subset. The two loss-based criteria perform worse due to uncertain, noisy samples, which blur the distinction between losses of clean and noisy samples in the training set and then further weakens the discrimination of the two loss-based criteria. Table 9 and Table 10 are the numerical versions of Figure 7.

### 5.6. The Effectiveness of Tripartition for the Other Methods

It is well worth testing whether the data divided by Tripartition can also help other methods. Then, we conduct a control experiment for the small-loss-based and GMM-based methods. Specifically, we replace their training data with the one partitioned by our Tripartition and apply their own learning strategies to verify if their performance would be improved. Firstly, we train two networks with different initializations (PreActResNet-18) on CIFAR-100 without label noise for 20 epochs. They achieve accuracies of 64.18% and 64.56%, respectively. Later on, networks are trained on CIFAR-100 with 30% instance-dependent noise and realistic noise. Take co-teaching on realistic noise as an example. In the experimental group, co-teaching applies its own learning strategy to the data divided by the small-loss criterion. In the control group, co-teaching uses the same learning strategy on the data divided by our Tripartition criterion. Two networks usually converge after 200 epochs. Their classification accuracies are listed in Table 11. For DivideMix, we follow the same protocol. Networks are trained by the data divided by the GMM criterion and the Tripartition criterion, respectively.

We can observe that our Tripartition criterion improves co-teaching and DivideMix by about 1.04% and 0.35% on instance-dependent noise, respectively, and improves them by about 0.99% and 0.05% on realistic noise. These results indicate that Tripartition selects a better clean subset than Small-loss and thus improves Co-teaching considerably. But we can see Tripartition is less helpful for DivideMix. The reason is that the uncertain subset divided by Tripartition is also learned by a semi-supervised strategy in DivideMix, in which clean samples are likely to be misused. The above experiments demonstrate that our Tripartition does reach a better partition of training data and can improve on other methods.

### 5.7. Ablation Study

To verify the effectiveness of our training strategies, the ablation study is conducted on CIFAR-100 using symmetric and realistic settings with varied noise ratios.

#### 5.7.1. The Training Strategy for the Uncertain Subset

The weight $\lambda_{u}$ is a key parameter for training the uncertain subset. Our motivation is to lower the harm of noisy labels; therefore, we test the $\lambda_{u}$ in the range of (0, 1]; and list results in Table 12. Obviously, $\lambda_{u}=1$ reaches the worst result because it preserves the harm. When $\lambda_{u}=0.3$, we have the best performance. However, $\lambda_{u}=0$ results in a worse result. The possible reason is that there are some uncertain clean samples, which are critical to the performance of the network. If they are excluded from training, the network can not learn valuable information from them.

#### 5.7.2. The Training Strategy for the Noisy Subset

To evaluate the effectiveness of the training strategy for the noisy subset, we test two strategies and report the results in Table 13. “Drop” means the noisy subset will be excluded from training. One can see that our semi-supervised strategy performs better because the given labels of noisy samples are replaced by the more reliable “co-guessed” labels from the predictions of two networks. This can minimize the harm of noisy labels and fully use these data. By contrast, dropping the mislabeled data performs worse because it does not use these data.

The $\lambda_{n}$ is a parameter to control the contribution of the noisy subset. Table 14 shows the detailed results with varied $\lambda_{n}$. One can see that Tripartite is not sensitive to $\lambda_{n}$, particularly with a low noise ratio and realistic noise. With symmetric noise with a ratio of 50%, the difference between the best and worst performances is less than 2%. In short, the dataset with a higher noise ratio and more complex data requires a larger $\lambda_{n}$ to learn sufficient information.

## 6. Conclusions

We present *Tripartite*—a novel partition method to prevent the model from misusing uncertain samples in training data by partitioning training data into three subsets: *uncertain*, *noisy*, and *clean*. Despite appearing to be an upgrade of the bipartition strategy, Tripartite addresses a core issue that is overlooked by binary approaches by carefully examining the nature of the distribution of noisy labeled data, especially on real-world label noise. Thus, it demonstrates a significant performance gain over SOTA methods on multiple types of label noise. Explorative analysis, such as the visualization of uncertain samples, the quality of data partition, and Tripartition for other methods, reveals the desirable trait, which is that Tripartition improves the quality of the clean subset and then the performance of models. We think Tripartite provides a new perspective on data partition in label noise learning.

While our proposed Tripartite demonstrates promising performance, it is pertinent to recognize some limitations. In an uncertain subset, some noisy samples may still exist. In future work, we will investigate other solutions to enhance the Tripartition strategy, in particular, more effectively utilizing the clean samples in the uncertain subset.

## Figures and Tables

**Figure 1 sensors-25-03369-f001:**
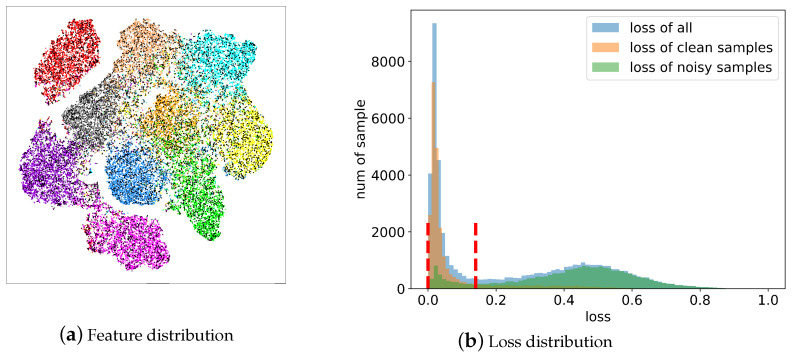
The feature distribution and loss distribution of the entire training set in CIFAR-10 with a realistic noise ratio of 50%. The testing model is a trained 18-layer PreAct Resnet [26] stopped at the early stage (40 epochs) using the DivideMix partition criterion [21]. (**a**) The feature distribution. The black dots represent the noisy samples. The colored dots represent the clean samples. Each color denotes a category. There are many uncertain samples distributed close to the classification boundary, in which some of them are grouped by the red curve. (**b**) The loss distribution. Some noisy samples also have small losses, as shown between two red dashed lines.

**Figure 2 sensors-25-03369-f002:**
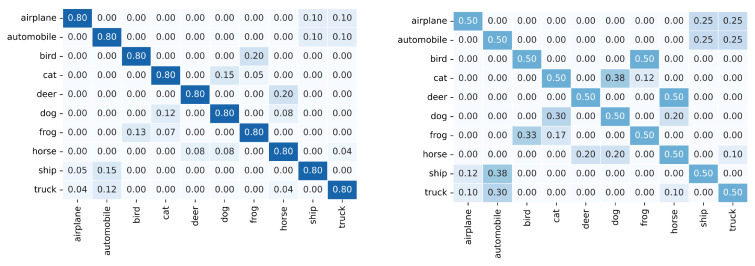
The generated noise transition matrices of CIFAR-10 with noise ratios of 20% and 50%.

**Figure 3 sensors-25-03369-f003:**
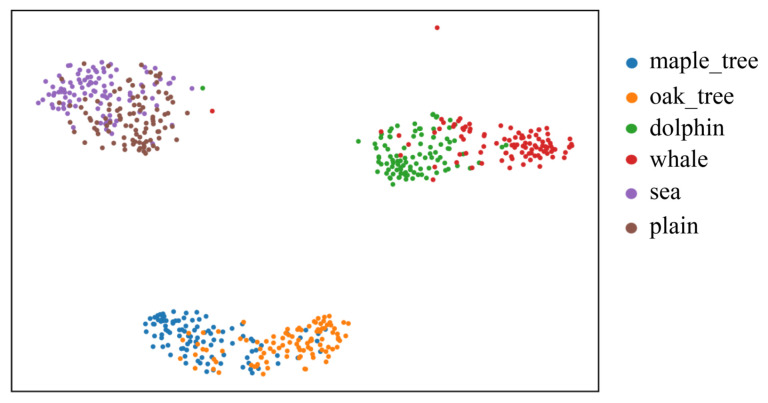
The visualization of the feature distributions of three similar class-pairs in CIFAR-100 using t-SNE [49].

**Figure 4 sensors-25-03369-f004:**
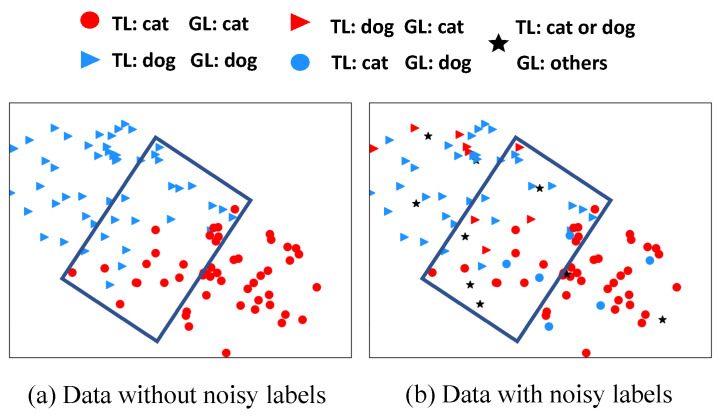
The distribution of 100 samples from categories “cat” and “dog” in the realistic CIFAR-10. Red and blue represent that given labels are cat and dog, respectively. Dot and triangle represent that the ground-truth labels are cat and dog, respectively. TL and GL denote the ground-truth label and the given label, respectively. Samples in the bounding box are considered as uncertain samples because they are close to the classification boundary. (**a**) The given labels are clean, equal to the ground-truth labels. (**b**) The given labels are noisy and differ from the ground-truth labels. Star represents a sample that is either a cat or dog but is given a noisy label from other classes.

**Figure 5 sensors-25-03369-f005:**
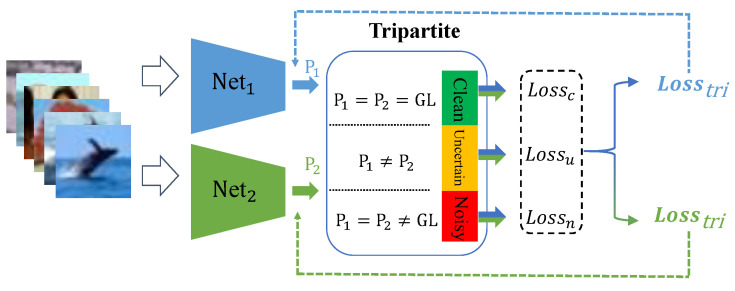
The logic of our Tripartite method. GL denotes the given label.

**Figure 6 sensors-25-03369-f006:**
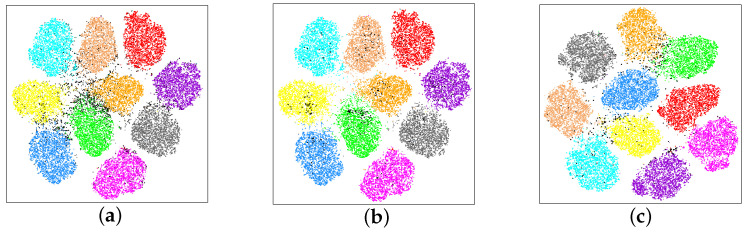
The networks are trained by our Tripartite solution in the later stage. To have a clearer visual effect, only clean and uncertain samples are visualized. (**a**) The distribution of uncertain samples (black dots) in the feature space of network 1. (**b**) A few uncertain samples distribute in the cluster in the feature space of network 1. (**c**) The uncertain samples shown in (**b**) distribute close to the classification boundary in the feature space of network 2.

**Figure 7 sensors-25-03369-f007:**
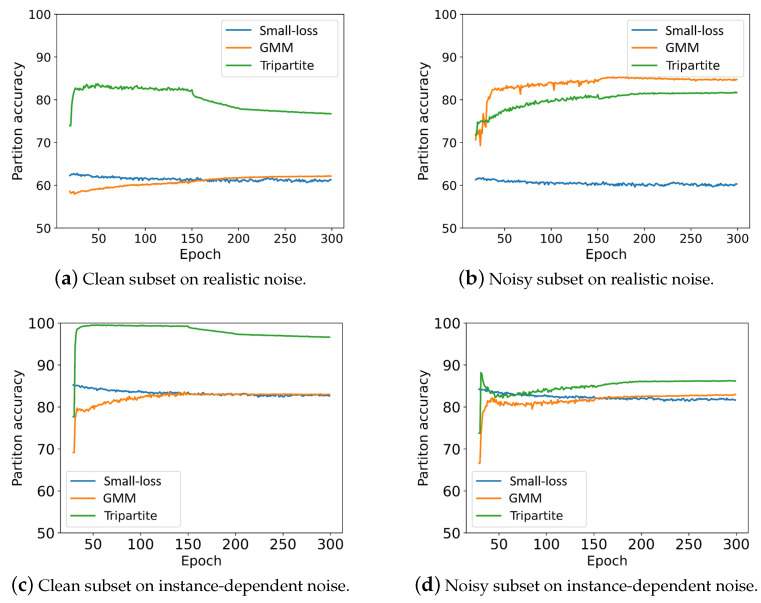
Comparison of three partition criteria on the CIFAR-100 dataset with 50% realistic label noise and 40% instance-dependent label noise. (**a**,**c**) The partition accuracy of clean samples in the clean subset against training epochs on two label noises. (**b**,**d**) The partition accuracy of noisy samples in the noisy subset against training epochs on two label noises.

**Table 1 sensors-25-03369-t001:** The top 10 similar class pairs in descending order of the cosine similarity from CIFAR-10. Table headers are bolded to enhances the readability of the table and makes the structural hierarchy clearer, Other forms are handled in the same way.

Similar Classes		Similarity
automobile	truck	0.075
automobile	ship	0.062
cat	dog	0.062
dog	horse	0.054
deer	horse	0.052
bird	frog	0.046
airplane	ship	0.044
horse	truck	0.025
cat	frog	0.021
airplane	truck	0.016

**Table 2 sensors-25-03369-t002:** The top 60 similar class pairs in descending order of the cosine similarity from CIFAR-100.

Similar Classes		Similarity	Similar Classes		Similarity	Similar Classes		Similarity
maple_tree	oak_tree	0.337	poppy	tulip	0.249	beaver	otter	0.222
dolphin	whale	0.312	bed	couch	0.246	lion	tiger	0.220
plain	sea	0.306	poppy	rose	0.245	clock	plate	0.220
bus	pickup_truck	0.301	snake	worm	0.245	pickup_truck	tank	0.219
girl	woman	0.298	bottle	can	0.242	flatfish	ray	0.217
bus	streetcar	0.296	apple	pear	0.242	plain	road	0.216
bicycle	motorcycle	0.275	dolphin	shark	0.237	chimpanzee	woman	0.216
baby	girl	0.273	baby	boy	0.232	shark	trout	0.215
maple_tree	willow_tree	0.270	poppy	sunflower	0.232	lawn_mower	tractor	0.214
castle	house	0.260	bowl	plate	0.231	house	streetcar	0.211
oak_tree	pine_tree	0.258	cloud	plain	0.231	leopard	lion	0.210
cloud	sea	0.257	hamster	mouse	0.230	otter	seal	0.210
oak_tree	willow_tree	0.255	chair	couch	0.230	beetle	spider	0.209
orchid	tulip	0.255	bowl	cup	0.229	bed	chair	0.209
beetle	cockroach	0.254	rose	tulip	0.229	orange	pear	0.209
streetcar	train	0.253	apple	orange	0.229	maple_tree	rose	0.207
leopard	tiger	0.253	television	wardrobe	0.227	pickup_truck	tractor	0.205
man	woman	0.251	keyboard	telephone	0.227	forest	willow_tree	0.205
orange	sweet_pepper	0.250	boy	girl	0.224	crab	lobster	0.205
apple	sweet_pepper	0.250	mouse	shrew	0.223	mountain	sea	0.204

**Table 3 sensors-25-03369-t003:** The detailed configurations of four benchmark datasets.

Datasets	# of Training	# of Test	# of Class	Size
CIFAR-10	50,000	10,000	10	32 × 32
CIFAR-100	50,000	10,000	100	32 × 32
Clothing1M	1,000,000	10,526	14	*
WebVision1.0	2,400,000	50,000	1000	*

**Table 4 sensors-25-03369-t004:** Accuracy comparisons on CIFAR-10 and CIFAR-100 with 20% and 50% symmetric noises. We run the open-source code of DSOS and report the result. The results of ELR+, Co-learning, PES, TCL+, and DLT are from their papers, and the rest of the results are from [21]. The best results are in bold, and the second-best results are underlined. * means that the methods did not have results provided.

	Dataset *Symmetric*	CIFAR-10		CIFAR-100	
Method		20%	50%	20%	50%
Standard CE		86.8	79.4	62.0	46.7
Bootstrap (2015) [18]		86.8	79.8	62.1	46.6
F-correction (2017) [16]		86.8	79.8	61.5	46.6
Co-teaching + (2019) [22]		89.5	85.7	65.6	51.8
P-correction (2019) [35]		92.4	89.1	69.4	57.5
Meta-Learning (2019) [7]		92.9	89.3	68.5	59.2
M-correction (2019) [60]		94.0	92.0	73.9	66.1
DivideMix (2020) [21]		96.1	94.6	77.3	74.6
ELR + (2020) [36]		95.8	94.8	77.6	73.6
Co-learning (2021) [61]		92.5	84.8	66.7	55.0
PES(semi) (2021) [40]		95.9	**95.1**	77.4	74.3
DSOS (2022) [42]		92.7	87.4	75.1	66.2
TCL + (2023) [39]		96.0	94.5	79.3	74.6
DLT (2023) [41]		96.3	*	77.1	*
Tripartite		**96.4**	**95.1**	**81.1**	**76.6**

**Table 5 sensors-25-03369-t005:** Accuracy comparisons on CIFAR-10 and CIFAR-100 with 20% and 40% instance-dependent noises. The result of InstanceGM is from [63], and the rest of the results are from PES [40]. The best results are in bold and the second-best results are underlined.

	Dataset *Inst*	CIFAR-10		CIFAR-100	
Method		20%	40%	20%	40%
Standard CE		87.5	78.9	56.8	48.2
DivideMix (2020) [21]		95.5	94.5	75.2	70.9
ELR + (2020) [36]		94.9	94.3	75.8	74.3
PES(semi) (2021) [40]		95.9	95.3	77.6	76.1
InstanceGM (2023) [63]		96.7	96.4	79.7	78.5
Tripartite		**96.8**	**96.6**	**80.1**	**78.8**

**Table 6 sensors-25-03369-t006:** Accuracy comparisons on CIFAR-10 and CIFAR-100 with 20%, 40%, and 50% realistic noises. We strictly implemented the competing methods according to their open codes and papers. The backbone is the Pre-ResNet18. The best results are in bold and the second-best results are underlined.

	Dataset *Realistic*	CIFAR-10			CIFAR-100		
Method		20%	40%	50%	20%	40%	50%
Co-teaching (2018) [20]		82.7	74.4	55.8	50.3	40.9	32.5
JoCoR (2020) [23]		82.2	68.7	55.8	49.7	35.1	29.1
DivideMix (2020) [21]		91.1	92.3	91.8	76.2	66.1	59.5
ELR + (2020) [36]		95.1	92.9	91.2	75.8	72.5	60.6
Co-learning (2021) [61]		91.6	75.5	61.7	68.5	58.5	50.2
PES(semi) (2021) [40]		95.9	93.9	91.1	76.3	73.2	66.7
DSOS (2022) [42]		92.5	89.0	81.1	75.3	64.8	52.6
Tripartite		**96.2**	**95.9**	**94.4**	**80.9**	**75.5**	**71.4**

**Table 7 sensors-25-03369-t007:** Accuracy comparison on Clothing1M. The results of JNPL, DSOS, and TCL are from [39,42,62]. The rest of the results are from [36]. The best results are in bold and the second-best results are underlined.

Clothing1M	
Methods	Acc.
Standard CE	69.21
F-correction (2017) [16]	69.84
Joint (2018) [19]	72.16
Meta-Learning (2019) [7]	73.47
P-correction (2019) [35]	73.49
DivideMix (2020) [21]	74.76
ELR + (2020) [36]	74.81
JNPL (2021) [62]	74.15
PES(semi) (2021) [40]	74.99
DSOS (2022) [42]	73.63
TCL (2023) [39]	74.80
Tripartite	**75.23**

**Table 8 sensors-25-03369-t008:** Top-1 (Top-5) accuracy comparison on the WebVision1.0 and ILSVRC12. The result of DSOS is from [42], the result of TCL is from [39], and the rest of the results are from [36]. The best results are in bold and the second-best results are underlined.

	Dataset	WebVision		ILSVRC12	
Method		Top-1	Top-5	Top-1	Top-5
F-correction (2017) [16]		61.12	82.68	57.36	82.36
Decoupling (2017) [64]		62.54	84.74	58.26	82.26
D2L (2018) [65]		62.68	84.00	57.80	81.36
MentorNet (2018) [38]		63.00	81.40	57.80	79.92
Co-teaching (2018) [20]		63.58	85.20	61.48	84.70
Iterative-CV (2019) [57]		65.24	85.34	61.60	84.98
DivideMix (2020) [21]		77.32	91.64	75.20	90.84
ELR+ (2020) [36]		77.78	91.68	70.29	89.76
DSOS (2022) [42]		77.76	92.04	74.36	90.80
TCL (2023) [39]		79.10	92.30	75.40	92.40
Tripartite		**80.12**	**93.88**	**76.40**	**93.20**

**Table 9 sensors-25-03369-t009:** The partition accuracy of Tripartition on CIFAR-100 with 50% realistic noise.

Epoch	Acc. (%)	Clean Subset		Noisy Subset		Uncertain Subset
		Partition Acc.	Total Nums	Partition Acc.	Total Nums	
20	46.47	73.94	12,896	71.75	9597	27,507
50	59.45	83.34	18,927	77.66	19,902	11,171
100	62.57	82.69	20,722	79.81	19,551	9727
150	64.77	82.31	21,634	81.16	19,085	9281
200	66.15	78.01	25,936	81.55	20,369	3695
250	65.99	77.22	26,501	81.51	20,537	2962
300	65.80	76.75	26,831	81.73	20,370	2799

**Table 10 sensors-25-03369-t010:** The partition accuracy of Tripartition on CIFAR-100 with 40% instance-dependent noise.

Epoch	Acc. (%)	Clean Subset		Noisy Subset		Uncertain Subset
		Partition Acc.	Total Nums	Partition Acc.	Total Nums	
30	49.42	78.15	21,954	77.71	3647	24,399
50	67.13	99.48	21,585	82.58	17,871	10,544
100	70.23	99.44	23,077	84.26	18,487	8436
150	71.57	99.23	23,836	85.09	18,492	7672
200	75.03	97.42	27,218	86.07	20,170	2612
250	75.70	96.93	27,554	86.14	20,335	2111
300	75.40	96.61	27,732	86.20	20,344	1924

**Table 11 sensors-25-03369-t011:** The effectiveness of Tripartition for other methods on CIFAR-100 with two label noises. Bold values indicate the best performance results in each comparison group.

Noisy Type	Method	Partition Criterion	Acc. %
Instance-dependent 30%	Co-teaching	Small-loss	71.30
		Tripartition	**72.34**
	DivideMix	GMM	75.94
		Tripartition	**76.29**
Realistic 30%	Co-teaching	Small-loss	71.04
		Tripartition	**72.03**
	DivideMix	GMM	76.44
		Tripartition	**76.49**

**Table 12 sensors-25-03369-t012:** Evaluation of λu in the training strategy for the uncertain subset on CIFAR-100. Bold values highlight the optimal performance for each noise condition with the corresponding best $\lambda_u$ parameter setting.

	$\lambda_{u}=$	1	0.5	0.3	0.1	0
Noise Type						
Symmetric	20%	78.11	79.95	**81.14**	80.60	78.84
	50%	67.80	75.26	**76.63**	75.04	72.53
Realistic	20%	77.53	80.01	**80.85**	80.00	77.63
	50%	67.78	70.07	**71.40**	70.99	64.6

**Table 13 sensors-25-03369-t013:** Evaluation of the training strategy for the noisy subset on CIFAR-100. Bold values indicate the best performance achieved by each training strategy under different noise conditions.

	Noise Type	Symmetric		Realistic	
Strategy		20%	50%	20%	50%
Drop		79.8	72.97	79.7	68.6
Semi-supervised		**81.14**	**76.63**	**80.85**	**71.40**

**Table 14 sensors-25-03369-t014:** Evaluation of λn in the training strategy for the noisy subset on CIFAR-100. Bold values highlight the optimal performance for each noise condition with the corresponding best $\lambda_n$ parameter setting.

	$\lambda_{n}=$	0.6	0.5	0.4	0.3	0.2	0.1
Noise Type							
Symmetric	20%	80.10	80.26	80.53	81.00	**81.14**	80.29
	50%	75.91	**76.63**	76.46	76.03	74.97	74.85
Realistic	20%	79.96	80.02	80.60	**80.85**	80.65	79.93
	50%	70.60	**71.40**	71.12	71.37	70.19	69.58

## Data Availability

The datasets used in this study are publicly available benchmark datasets commonly used in machine learning research. The specific datasets and their sources are detailed in Section 4 of the manuscript. The code and experimental results supporting the conclusions of this article will be made available by the authors upon reasonable request.

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
