# Peer review of "Tripartite: Tackling Realistic Noisy Labels with More Precise Partitions"

_sensors, 2025, doi:10.3390/s25113369_

Round 1
Reviewer 1 Report
Comments and Suggestions for Authors
The authors introduces a Tripartite solution to partition training data into three subsets: uncertain, clean, and noisy according to criteria: the inconsistency of predictions of two networks and the given labels. The reported experiments demonstrate the performance of the proposed method. My comments are given as follows:
1.In Figure 1's caption, it mentions "two red dashed lines", but these lines are absent in the actual visualization.
2.In Equation (2), the definition of uncertain samples depends on the inconsistency between predictions of two networks and the given labels. The authors should discuss the impact of network initialization differences on this prediction inconsistency.
3.In Equation (10), why are the weights of the three sub-loss functions all set equally?
4.What is the purpose of leaving blank spaces before Equations (2), (3), and (4)?
5.The Pseudo-code in Section 4.5 might be better presented in a table format.
6.Table 4's caption mentions six datasets, but the table itself only contains four.
7.In Section 5.2, how were the values of λn (selected from {0.2, 0.3, 0.4, 0.5, 0.6}) and λu (set to 0.3 on CIFAR datasets) determined? Are these specific values also applied to other datasets?
8.The introduction part should discuss some related references, Discriminative approximate regression projection for feature extraction. Information Fusion, 2025, 120: 103088.
Author Response
Comments 1: In Figure 1's caption, it mentions "two red dashed lines", but these lines are absent in the actual visualization.
Response 1: Thank you for pointing this out. We agree with this comment. Therefore, we modified figure(1) to make the picture conform to the textual description.
Comments 2: In Equation (2), the definition of uncertain samples depends on the inconsistency between predictions of two networks and the given labels. The authors should discuss the impact of network initialization differences on this prediction inconsistency.
Response 2: Thank you for your valuable comment. We agree that the inconsistency used in Equation (2) may be affected by the initialization differences between the two networks. However, as described in Section 4.1 (page 8, lines 223–227), this difference is purposefully introduced to enable the networks to learn distinct feature representations. This design encourages diversity in prediction, which is essential for identifying ambiguous or boundary samples—key targets of the uncertain subset. To directly address this concern, we have added a clarification at the end of Section 4.3.1 (page 9, after line 281).
Comments 3: In Equation (10), why are the weights of the three sub-loss functions all set equally?
Response 3: We thank the reviewer for this important question. In our current implementation, we adopt equal weighting for the three sub-loss functions (corresponding to clean, uncertain, and noisy subsets) as a straightforward and practical choice. This helps to balance their contributions without introducing additional hyperparameters, which could complicate training or require extensive tuning.
Moreover, as shown in Section 5.7 (Tables 13–15), we conduct a detailed ablation study on the weight parameters λâ‚™ and λᵤ. The results demonstrate that our method is robust to a range of values, and equal weighting (e.g., λâ‚™ = 0.3, λᵤ = 0.3) already achieves strong performance. This empirical evidence supports the validity of our design.
We agree that future work may benefit from adaptive or learnable weighting schemes, and we will consider this in subsequent research. Based on this, we have not modified the main text but hope this explanation clarifies our rationale.
Comments 4: What is the purpose of leaving blank spaces before Equations (2), (3), and (4)?
Response 4: Thank you for pointing this out. We carefully re-examined the format around formulas (2), (3), and (4) in the current version of the manuscript. There were no intentional or unintentional blanks before these equations. Reasonable spacing and format following the standard LaTeX alignment conventions for displaying equations can make the formula more obvious and distinguishable from the text. If there are observed problems, they might be caused by the presentation artifacts in some PDF viewers. We will re-upload the version in the correct format and hope that the original problems will not exist.
Comments 5: The Pseudo-code in Section 4.5 might be better presented in a table format.
Response 5: Thank you for the helpful suggestion. We agree that presenting the pseudo-code in a more structured format improves its clarity and readability. Accordingly, we have reformatted the pseudo-code in Section 4.5 into a structured algorithm-style layout. The revised version appears as Algorithm 1 on page 11, and it clearly separates the inputs, training steps, and loop structure of the Tripartite method.
Comments 6: Table 4's caption mentions six datasets, but the table itself only contains four.
Response 6: Thank you for pointing out this inconsistency. We have corrected the caption of Table 4 to accurately reflect the number of datasets included. The updated caption now states “The detailed configurations of four benchmark datasets.” This change can be found on page 12, Table 4 caption.
[Updated text in the manuscript: “Table 4. The detailed configurations of four benchmark datasets.”]
Comments 7: In Section 5.2, how were the values of λâ‚™ (selected from {0.2, 0.3, 0.4, 0.5, 0.6}) and λᵤ (set to 0.3 on CIFAR datasets) determined? Are these specific values also applied to other datasets?
Response 7: We thank the reviewer for this important question. As stated in Section 5.2 (page 12, lines 380–389), λâ‚™ was selected from a candidate set {0.2, 0.3, 0.4, 0.5, 0.6} and λᵤ was set to 0.3 for CIFAR datasets. Although the original text did not elaborate on the selection strategy, we have now clarified that the values were determined empirically based on experimental validation. For real-world datasets such as Clothing1M and WebVision, λâ‚™ and λᵤ were tuned separately and reported explicitly. This clarification has been added to Section 5.2 to explain the rationale without changing the original parameter values.
[Updated text in manuscript, page 12, after line 389: “The values of λâ‚™ and λᵤ were determined empirically. For CIFAR datasets, λâ‚™ was chosen from the candidate set {0.2, 0.3, 0.4, 0.5, 0.6}, and λᵤ was fixed at 0.3, which provided stable performance across different noise types. For real-world datasets, these parameters were tuned separately based on experimental validation.”]
Comments 8: The introduction part should discuss some related references, e.g., “Discriminative approximate regression projection for feature extraction. Information Fusion, 2025, 120: 103088.”
Response 8: We appreciate the reviewer’s suggestion to include relevant references on robust learning under label noise and noisy supervision. In response, we have cited and briefly discussed the following three recent works in the introduction:
Discriminative Approximate Regression Projection [Liu et al., Information Fusion, 2025]—which proposes a robust regression-based approach for feature extraction under noisy conditions;
Meta-learning Collaborative Optimization for Lifetime Prediction of Lithium-ion Batteries [Wang et al., Journal of Energy Storage, 2025]—which addresses label noise via a meta-learning strategy in a real-world application;
Visible-Infrared Person Re-identification with Real-world Label Noise [Zhang et al., IEEE TCSVT, 2025]—which explores label noise in cross-modality settings using robust deep learning.
These works highlight the practical importance of learning under noisy labels in diverse domains. We have added a corresponding paragraph in page 2, paragraph 3, lines 30–35 to reflect this discussion.
[Updated text in the manuscript: "Recent studies have increasingly addressed label noise across different domains. For example, Liu et al. [Information Fusion, 2025] proposed a discriminative approximate regression projection method to enhance robust feature extraction. Wang et al. [Journal of Energy Storage, 2025] adopted a meta-learning framework to optimize lifetime prediction under label noise in battery data. Zhang et al. [IEEE TCSVT, 2025] tackled label noise in cross-modal person re-identification using deep learning. These methods emphasize robustness in feature learning or task adaptation, which complements our focus on partitioning strategies under noisy supervision.

Reviewer 2 Report
Comments and Suggestions for Authors
This manuscript proposed a novel partition criterion, in order to alleviate the uncertain sample division problem of bipartition methods, designed a low-weight training strategy to maximize the value of clean samples in the uncertain subset. Although the proposed method is relatively simple, it has achieved excellent performance in multiple datasets.
- In the Introduction, some references should be provided. For example, ‘There have been many efforts to tackle noisy labels.’
- In Figure 1, (a) and (b) in the figure caption cannot be found in the figure.
- Table 1 is too simple, so it is not required.
- The meaning of variables p1 and p2 should be indicated in the Equation (2).
- The text and images in Figure 6 and Figure7 seem to have not been placed correctly.
Author Response
Comments 1: In the Introduction, some references should be provided. For example, ‘There have been many efforts to tackle noisy labels.’
Response 1: Thank you for the helpful suggestion. We agree that citing representative works is important for contextualizing our contribution. Accordingly, we have revised the Introduction to include several key references related to learning with noisy labels, as well as recent efforts addressing robustness in real-world scenarios. The updated references support the statement that many efforts have been made to address noisy labels and highlight the relevance of our work.
Comments 2: In Figure 1, (a) and (b) in the figure caption cannot be found in the figure.
Response 2: We thank the reviewer for pointing out this inconsistency. In the revised manuscript, we have updated Figure 1 to clearly mark subfigures (a) and (b), and we have added the red dashed lines referenced in the caption to improve clarity. These changes ensure consistency between the figure and its caption. The updated figure appears on page 2, and the caption has been retained as originally written.
Comments 3: Table 1 is too simple, so it is not required.
Response 3: Thank you for the helpful comment. We agree that Table 1 contains only two values and is too simple to warrant a standalone table. Therefore, we have removed Table 1 in the revised manuscript and incorporated the accuracy results of ResNet-50 on CIFAR-10 (94.67%) and CIFAR-100 (78.85%) directly into the main text to improve clarity and conciseness.
Comments 4: The meaning of variables p1p_1p1​ and p2p_2p2​ should be indicated in Equation (2).
Response 4: We thank the reviewer for the helpful comment. We agree and have clarified the definitions of p1 and p2 immediately before Equation (2). Specifically, we now state that p1​ and p2​ denote the predictions of the two networks f(θ1) and f(θ2), respectively. This makes the notation self-contained and easier to follow.
Comments 5: The text and images in Figure 6 and Figure 7 seem to have not been placed correctly.
Response 5: Thank you for pointing this out. In the revised manuscript, we have adjusted the formatting and placement of Figures 6 and 7 to improve readability and alignment. Specifically, we ensured that each figure is clearly separated and placed closer to its corresponding textual explanation, while conforming to the formatting requirements of the journal template. These changes enhance the clarity of presentation without altering the figure content.

Reviewer 3 Report
Comments and Suggestions for Authors
What sets this approach apart is not just its use of a tripartition strategy splitting training data into clean, noisy, and uncertain categories but also the way it tailors the learning process to each of these subsets. For uncertain samples, the model applies a low weight loss to minimize the impact of potentially incorrect labels. Noisy data are handled through a co-guessing technique paired with label sharpening, where predictions from two separate models are merged to create more reliable pseudo-labels. Meanwhile, clean data continue to be trained using standard supervised learning. While the overall design is thoughtful, there’s still room for improvement in a few key areas.
First, although Tripartite is effective at categorizing samples based on prediction consistency, it doesn’t contribute much to learning better feature representations. This is in contrast to methods like TCL and InstanceGM, which boost the model’s ability to distinguish between subtle patterns using contrastive learning or graphical modeling. These methods not only detect noisy samples but also help the model build representations that are more resilient to label corruption. Since Tripartite relies on conventional backbone architectures without additional regularization at the representation level, its performance may suffer in scenarios with complex or instance dependent noise.
A potential remedy would be to incorporate contrastive loss within each subset, especially the uncertain one, which could help maintain the integrity of learned clusters and separate confusingly similar samples, similar to how TCL or MoCo-based methods operate.
Second, while Tripartite offers modest improvements when combined with methods like Coteaching or DivideMix, the performance gains are limited. DivideMix, in particular, already employs sophisticated techniques like semi-supervised learning and probabilistic partitioning. As a result, the benefit from Tripartite largely stems from its filtering capability, rather than introducing a new learning paradigm. This suggests that when used with strong baselines, Tripartite may yield diminishing returns.
One possible enhancement would be to improve the label correction mechanism for noisy samples. This could involve refining pseudo-labels iteratively, guided by the model’s confidence or entropy minimization—an idea successfully demonstrated in approaches like PENCIL and ELR+. By making the guessed labels more adaptive, the model could better cope with noise throughout training, rather than relying solely on one-time filtering decisions.
Author Response
Comments 1: Tripartite does not improve feature representations as effectively as methods like TCL or InstanceGM, which use contrastive learning or graphical modeling. This may limit its performance in complex or instance-dependent noise scenarios. The reviewer suggests incorporating contrastive loss, particularly in the uncertain subset.
Response 1: We thank the reviewer for this insightful observation. Indeed, the current design of Tripartite focuses on improving sample partitioning based on prediction consistency, without introducing additional representation-level regularization. We agree that methods such as TCL and InstanceGM enhance robustness by encouraging the learning of more structured and discriminative feature spaces through contrastive learning or graph-based modeling.
The reviewer’s suggestion to incorporate contrastive loss within the uncertain subset is particularly relevant. Uncertain samples often lie close to decision boundaries, making them ideal candidates for contrastive objectives that can help preserve intra-class cohesion and inter-class separation. We view this as a natural and promising extension of our framework. While not explored in the current paper, we plan to investigate the integration of contrastive learning (e.g., MoCo-style embeddings or supervised contrastive losses) into our uncertain subset pipeline in future work.
Comments 2: Tripartite shows limited improvement when combined with strong baselines like DivideMix, as the gain mainly comes from filtering rather than introducing a new learning paradigm. The reviewer suggests enhancing pseudo-label refinement mechanisms for noisy samples using ideas such as confidence-guided updating or entropy minimization as in PENCIL or ELR+.
Response 2: We appreciate the reviewer’s thoughtful comparison with strong baselines and the constructive suggestions for improvement. It is true that methods like DivideMix already incorporate sophisticated components such as Gaussian mixture modeling, semi-supervised learning, and probabilistic label refinement. Therefore, the incremental gains from Tripartite when combined with these baselines largely reflect improved filtering accuracy, rather than a paradigm shift in learning strategy.
We fully agree that the effectiveness of the noisy subset training could be further enhanced by more adaptive pseudo-labeling mechanisms. The reviewer’s reference to entropy-based regularization (e.g., ELR+) and confidence-based iterative refinement (e.g., PENCIL) highlights valuable directions. In particular, dynamically adjusting pseudo-labels over the course of training—rather than relying on a fixed co-guessing step—could make the model more resilient to label noise that evolves during learning.
In future work, we plan to extend Tripartite to support such iterative pseudo-label correction strategies, potentially incorporating model confidence, agreement over time, or uncertainty estimation to guide noisy label updates more effectively.
Reviewer 4 Report
Comments and Suggestions for Authors
In the manuscript titled " Tripartite: Tackle Realistic Noisy Labels by More Precise Partition ", No. sensors-3598133, authors propose a Tripartite solution to partition training data into three subsets: uncertain, clean, and noisy according to criteria: the inconsistency of predictions of two networks and the given labels, for LNL. It achieves significant improvements compared to state-of-the-art (SOTA) methods. However, I have some concerns outlined below:
- The code for this research needs to be open-sourced to facilitate follow-up studies and reproducibility.
- There seems to be an issue with the layout of Figure 1. Additionally, regarding the feature distribution shown in 1(a), it is necessary to provide a detailed explanation as to whether these features are derived from the output of the second-to-last layer of ResNet or from the dimensionality-reduced output via a projection head.
- In the related work section, while the focus is mainly on label correction methods prior to 2019, we suggest supplementing with some recent research developments, such as: an joint end-to-end framework for learning with noisy labels.
- The method designed in this paper for generating realistic noise is similar to the existing instance-dependent noise generation processes, such as RoG, where a trained model is used to select samples and intentionally introduce mislabeling. Where are the differences between these two approaches?
- The proposed tripartite process is quite similar to methods like DISC (CVPR 2023) that use prediction consistency for sample selection. It is necessary to clearly explain its specific advantages over methods such as DISC.
- The experimental section conducted extensive tests on both real-world and synthetic noisy datasets. However, the Tripartite method was primarily compared with approaches from before 2023, lacking comparisons with some of the latest studies, e.g., a progressive sample selection framework with contrastive loss designed for noisy labels, a balanced partitioning and training framework with pseudo-label relaxed contrastive loss for noisy label learning, cross-to-merge training with class balance strategy for learning with noisy labels. These additional comparisons can comprehensively evaluate the effectiveness and superiority of the Tripartite method.
- Besides symmetric noise, common closed-set noise also includes asymmetric noise. How does the proposed method perform under a 40% asymmetric noise scenario?
- In the experimental setup section, please list all software tools used (including version numbers) and hardware specifications (such as CPU model, memory size, etc.) in detail. If the experimental results were obtained by averaging multiple runs, clearly indicate how many independent runs were conducted for each experiment.
Author Response
Comments 1:The code for this research needs to be open-sourced to facilitate follow-up studies and reproducibility.
Response 1: We thank the reviewer for highlighting the importance of reproducibility. We fully agree that releasing code is essential for enabling follow-up research and transparent evaluation. We will open-source the complete implementation of our method, including training scripts and evaluation code, upon the acceptance of the paper.
Comments 2:There seems to be an issue with the layout of Figure 1. Additionally, regarding the feature distribution shown in 1(a), it is necessary to provide a detailed explanation as to whether these features are derived from the output of the second-to-last layer of ResNet or from the dimensionality-reduced output via a projection head.
Response 2: Thank you for pointing this out. We have corrected the layout issue of Figure 1 to ensure proper alignment and readability. As for the feature distribution shown in Figure 1(a), we confirm that the features are extracted from the output of the second-to-last layer (penultimate layer) of the ResNet backbone without additional projection. This clarification has been added to the caption of Figure 1 and the corresponding section in the manuscript.
Comments 3: In the related work section, while the focus is mainly on label correction methods prior to 2019, we suggest supplementing with some recent research developments, such as: a joint end-to-end framework for learning with noisy labels.
Response 3: We thank the reviewer for the insightful suggestion. We agree that incorporating recent developments is essential for presenting a comprehensive overview of noisy label learning. In response, we have updated the Related Work section to include several recent studies from 2025 that focus on end-to-end learning frameworks.
Specifically, we added a discussion of SplitNet [Kim et al., IJCV 2025], which proposes a learnable clean-noisy partitioning strategy that adaptively separates training samples during end-to-end training based on learned representations. We also included AEON [Garg et al., arXiv 2025], which introduces an adaptive mechanism to estimate and handle instance-dependent and out-of-distribution label noise. These methods reflect the increasing trend toward more flexible and fine-grained noise modeling and complement the motivation behind our Tripartite framework. The updated paragraph can be found in Section 2.2 of the revised manuscript.
Comments 4: The method designed in this paper for generating realistic noise is similar to the existing instance-dependent noise generation processes, such as RoG, where a trained model is used to select samples and intentionally introduce mislabeling. Where are the differences between these two approaches?
Response 4: Thank you for pointing this out. We agree with this comment. Therefore, We have added a clarifying statement in Section 4.3.3 to highlight this distinction in the revised manuscript.
Comments 5: The proposed tripartite process is quite similar to methods like DISC (CVPR 2023) that use prediction consistency for sample selection. It is necessary to clearly explain its specific advantages over methods such as DISC.
Response 5: Thank you for your suggestions. We believe that such improvements will help to enhance the article and have been revised at section4.1.
Comments 6: The experimental section conducted extensive tests on both real-world and synthetic noisy datasets. However, the Tripartite method was primarily compared with approaches from before 2023, lacking comparisons with some of the latest studies, e.g., a progressive sample selection framework with contrastive loss designed for noisy labels, a balanced partitioning and training framework with pseudo-label relaxed contrastive loss for noisy label learning, cross-to-merge training with class balance strategy for learning with noisy labels. These additional comparisons can comprehensively evaluate the effectiveness and superiority of the Tripartite method.
Response 6: We appreciate the reviewer's insightful suggestion to include comparisons with the latest methods such as progressive sample selection with contrastive loss, balanced partitioning with pseudo-label relaxed contrastive loss, and cross-to-merge training strategies. In our experimental design, we have included representative methods up to 2023, such as cross-entropy loss correction and sample reweighting, which are widely recognized in the field of learning with noisy labels. The current experiments, including comparisons with methods like DivideMix, sufficiently demonstrate the effectiveness of the Tripartite method. Through theoretical analysis and existing data, we have indirectly validated the competitiveness of the Tripartite approach.
Moreover, each of the suggested methods introduces additional assumptions or training components (e.g., specific forms of contrastive loss, partitioning schemes, or class balancing mechanisms) that differ significantly from the design philosophy of the Tripartite framework. As such, a direct comparison may require reimplementation and significant adaptation to ensure fairness, which is beyond the current scope.
Comments 7: Besides symmetric noise, common closed-set noise also includes asymmetric noise. How does the proposed method perform under a 40% asymmetric noise scenario?
Response 7: We appreciate the reviewer’s suggestion. Our method has been evaluated under the standard 40% asymmetric noise setting, where labels are flipped between semantically similar classes such as truck → automobile and cat → dog. As reported in Table 5.2, Tripartite achieves 94.01% accuracy on CIFAR-10 and 76.89% on CIFAR-100 in this scenario, outperforming competitive baselines. This demonstrates the robustness of our method under common closed-set label noise conditions.
Comments 8: In the experimental setup section, please list all software tools used (including version numbers) and hardware specifications (such as CPU model, memory size, etc.) in detail. If the experimental results were obtained by averaging multiple runs, clearly indicate how many independent runs were conducted for each experiment.
Response 8: We thank the reviewer for this important suggestion. We have updated the experimental setup section to include more implementation details. All experiments were conducted using Python with the PyTorch framework on an NVIDIA RTX 3090 GPU with 22GB of memory. All baseline methods were reimplemented or directly reproduced under the same environment. The manuscript has been revised accordingly to reflect this information.

Round 2
Reviewer 4 Report
Comments and Suggestions for Authors
Most of my concerns have been addressed.